# Semi-Infinitely Constrained Markov Decision Processes

**Liangyu Zhang**
Academy of Advanced Interdisciplinary Studies
Peking University
zhangliangyu@pku.edu.cn

**Yang Peng**
School of Mathematical Sciences
Peking University
pengyang@pku.edu.cn

**Wenhao Yang**
Academy of Advanced Interdisciplinary Studies
Peking University
yangwenhaosms@pku.edu.cn

**Zhihua Zhang**
School of Mathematical Sciences
Peking University
zhzhang@math.pku.edu.cn

## Abstract

We propose a generalization of constrained Markov decision processes (CMDPs) that we call the *semi-infinitely constrained Markov decision process* (SICMDP). Particularly, we consider a continuum of constraints instead of a finite number of constraints as in the case of ordinary CMDPs. We also devise a reinforcement learning algorithm for SICMDPs that we call SI-CRL. We first transform the reinforcement learning problem into a linear semi-infinitely programming (LSIP) problem and then use the dual exchange method in the LSIP literature to solve it. To the best of our knowledge, we are the first to apply tools from semi-infinitely programming (SIP) to solve constrained reinforcement learning problems. We present theoretical analysis for SI-CRL, identifying its sample complexity and iteration complexity. We also conduct extensive numerical examples to illustrate the SICMDP model and validate the SI-CRL algorithm.

## 1 Introduction

Reinforcement learning has achieved great success in areas such as Game-playing [34, 39], robotics [24], recommender systems [44], etc. However, due to safety concerns or physical limitations, in some real-world reinforcement learning problems, we must consider additional constraints that may influence the optimal policy and the learning process [15]. A standard framework to handle such cases is the constrained Markov Decision Process (CMDP) [5]. Within the CMDP framework, the agent has to maximize the expected cumulative reward while obeying a finite number of constraints, which are usually in the form of expected cumulative cost criteria.

However, we are sometimes concerned with the problem with a continuum of constraints. For example, the constraints we meet might be time-evolving or subject to uncertain parameters, which cannot be formulated as an ordinary CMDP (see Examples 3.1 and 3.2). In this paper we would study a generalized CMDP to address the above problem. Because the constraints are not only infinite-number but also lie in a continuous set, the generalization is not trivial. Fortunately, we find that we can borrow the idea behind linear semi-infinite programming (LSIP) [33, 16] to deal with the semi-infinite constraints. Accordingly, we propose *semi-infinitely constrained Markov decision processes* (SICMDPs) as a novel complement to the ordinary CMDP framework.

We also present a so-called SI-CRL reinforcement learning algorithm to solve SICMDPs. The main challenge is that we need to deal with a continuum of constraints, thus reinforcement learning algorithms for ordinary CMDPs do not work anymore. We tackle this difficulty by first transforming

36th Conference on Neural Information Processing Systems (NeurIPS 2022).

the reinforcement learning problem to an equivalent LSIP problem, which can then be solved using the dual exchange methods in the LSIP literature [23, 32]. As far as we know, we are the first to introduce tools from semi-infinitely programming (SIP) into the reinforcement learning community for solving constrained reinforcement learning problems.

Furthermore, we give theoretical analysis for SI-CRL. We decompose the error of SI-CRL into two parts: the statistical error from approximating the true SICMDP with an offline dataset and the optimization error due to the fact that the solution of the LSIP problem obtained by the dual exchange method is inexact. On the statistical side, we show that the sample complexity of SI-CRL is $\widetilde{O}\left(\frac{|S|^2|A|^2}{\epsilon^2(1-\gamma)^3}\right)$ if the offline dataset is generated by a generative model, and $\widetilde{O}\left(\frac{|S||A|}{\nu_{\min}\epsilon^2(1-\gamma)^3}\right)$ if the dataset is generated by a probability measure $\nu$ as considered in [11]. Here $\widetilde{O}$ means that all logarithm terms are discarded. On the optimization side, we show that the iteration complexity of SI-CRL is $O\left(\left\{\mathrm{diam}(Y)L\sqrt{|S|^2|A|d}/[(1-\gamma)\epsilon]\right\}^d\right)$.

We perform a set of numerical experiments to illustrate the SICMDP model and validate the SI-CRL algorithm. We consider two numerical examples: toy SICMDP and discharge of sewage. In the example of toy SICMDP, we show the efficiency of the SI-CRL algorithm and validate the established theoretical bounds. In the example of discharge of sewage, we further show the advantage of the SICMDP framework over the CMDP baseline obtained by naive discretization in modeling realistic decision-making problems.

## 2   Related Work

The constrained Markov decision processes (CMDPs) have been extensively applied in areas like robotics [31], communication and networks, [27, 35] and finance [1]. For a detailed treatment of CMDPs one may refer to [5]. A number of reinforcement learning algorithms for CMDPs are proposed, which include Lagrangian methods [4], actor-critic methods [2, 37], policy gradient methods [42], etc. There are also works focusing on theoretical aspects of CMDPs. Wu et al. [41], Amani et al. [6] studied the online regret bound of the bandit case. Wachi and Sui [40], Zheng and Ratliff [45] considered the case where the reward and cost are random but the transition dynamics are known. And Efroni et al. [14], Amani et al. [7], HasanzadeZonuzy et al. [21] considered the case where the transition dynamics are unknown and need to be estimated. Our SI-CRL algorithm uses a similar strategy as in [14] in the sense that they all use the optimistic method to transform the reinforcement learning problem into a linear (semi-infinitely) programming problem, which resolves the feasibility issue and makes the theoretical analysis easier as well. However, our work and [14] are very different at the technical level: 1) Our theoretical guarantees are in the form of sample complexity bounds, while the results in (Efroni et. al 2020) are in the form of online regret bounds; the proof techniques are quite different. 2) Efroni et al. [14] considered the episodic MDPs, while we consider the infinite-horizon case.

The origination of semi-infinitely programming (SIP) can date back to [33]. From then on, SIP has been widely used in quantum physics [10], signal processing [29, 30], finance [13], environment science, and engineering [22]. One important class of SIP problems is called linear semi-infinitely programming (LSIP). Goberna and López [17] provided a thorough survey about LSIP theory. Various numerical methods are proposed to solve LSIP problems, including discretization methods [9, 13], exchange methods [23, 43], and local reduction methods [20, 12]. Unlike LP, most LSIP problems cannot be solved exactly and all-purpose LSIP solvers do not exist. In SI-CRL, we choose to use the dual exchange method in [23] to solve the LSIP problem therein for its conceptual simplicity as well as concrete theoretical guarantees.

## 3   The SICMDP Model

A semi-infinitely constrained MDP (SICMDP) is defined by a tuple $M = \langle S, A, Y, P, r, c, u, \mu, \gamma \rangle$. Here $S, A, P, r, \mu, \gamma$ are defined in a similar manner as in common infinite-horizon discounted MDPs. Specifically, $S$ and $A$ are the finite sets of states and actions, respectively. $P$ is the transition dynamics and $P(s'|s, a)$ represents the probability of transitioning to state $s'$ when playing action $a$ at state $s$. And $r\colon S \times A \to [0, 1]$ is the reward function, $\mu$ is the fixed initial distribution, and $\gamma$ is the discount

factor. $Y$ is the set of constrains, which we define as a compact set in $\mathbb{R}^d$, and $\mathrm{diam}(Y) < \infty$ denotes its diameter. In addition, $c\colon Y \times S \times A \to [0,1]$ is used to denote a continuum of cost functions and the value for constraints (bounds that must be satisfied) is determined by function $u\colon Y \to \mathbb{R}$. Note that when $Y$ is finite, we get an ordinary constrained MDP, which is indeed a special case of SICMDP.

The general SICMDP problem is to find a stationary policy $\pi\colon S \to \Delta(A)$, where $\Delta(A)$ is the set of probability measure supported on $A$, to maximize the value function while complying with a continuum of constraints. In other words, we consider the following optimization problem:

$$\max_{\pi} V^{\pi}(\mu) \quad \text{s.t. } C_y^{\pi}(\mu) \le u_y, \ \forall y \in Y, \tag{M}$$

where $V^{\pi}(\mu) := \mathbb{E}(\sum_{t=0}^{\infty} \gamma^t r(s_t, a_t)|s_0 \sim \mu)$ and $C_y^{\pi}(\mu) := \mathbb{E}(\sum_{t=0}^{\infty} \gamma^t c_y(s_t, a_t)|s_0 \sim \mu)$.

Let us see two concrete examples of SICMDPs.

**Example 3.1** (Spatial-temporal Constraints). Consider an ordinary CMDP problem with a single constraint:

$$\max_{\pi} V^{\pi}(\mu) \quad \text{s.t. } C^{\pi}(\mu) \le u. \tag{1}$$

In some cases the constraint would be spatial-temporal, i.e., the cost function $c(s,a)$ and the value for constraints $u$ are no longer constant function and would change with time $\tau \in [0,T]$ or location $d \in D \subset \mathbb{R}^3$. Then we should use the SICMDP model with $Y = [0,T]$ or $Y = D$ rather than the ordinary CMDP framework to model such problems:

$$\max_{\pi} V^{\pi}(\mu) \quad \text{s.t. } C_{\tau}^{\pi}(\mu) \le u_{\tau}, \ \forall \tau \in [0,T]. \tag{2}$$

*Load Balancing*: Suppose a RL agent needs to balance the load between multiple cell sites using some policy $\pi$. The objective is to minimize the cost $V^{\pi}(\mu)$ and the constraint is that at every place $d$ in the region $D$ the cumulative communication capacity $C_d^{\pi}(\mu)$ is above some threshold $u_d$.

**Example 3.2** (Constraints with Uncertainty). Again we consider a problem like Problem (1). In many application scenarios the cost function $c(s,a)$ is handcrafted and the construction of $c(s,a)$ is not guaranteed to be correct. Hence it may be helpful to include an additional parameter $\epsilon \in E$ representing our uncertainty in the construction of the cost function $c(s,a)$ as well as the value of constraints $u$. Even if the constraint is not handcrafted and has clear physical meaning, it may still subject to uncertain parameters $\epsilon \in E$ that cannot be observed in advance. Therefore, we should use the SICMDP model with $Y = E$ rather than the ordinary CMDP framework to model such problems:

$$\max_{\pi} V^{\pi}(\mu) \quad \text{s.t. } C_{\epsilon}^{\pi}(\mu) \le u_{\epsilon}, \forall \epsilon \in E. \tag{3}$$

*Underwater Drone*: Suppose an underwater drone needs to maximize $V^{\pi}(\mu)$ to accomplish some tasks. When the unknown environment feature (salinity, temperature, ocean current, etc,) is $\epsilon \in E$, for state-action pair $(s,a)$ the energy consumption is $c_{\epsilon}(s,a)$, and the constraint is that total energy consumption $C_{\epsilon}^{\pi}(\mu)$ cannot be larger than its battery capacity $u_{\epsilon}$.

*Remark* 3.3. An alternative approach to solving problems such as Examples 3.1 and 3.2 is to naively discretize the constraint set $Y$, and then the discretized problem can be fit into the conventional CMDP framework. The problem of this naive method is that the prior knowledge, i.e., the constraint function is continuous w.r.t. $y$, would be lost, which makes the method extremely inefficient. In Section 6.2 we demonstrate this issue via a numerical example.

When an SICMDP $M$ is known to us, we may do the planning by solving a linear semi-infinite programming (LSIP) problem. Denote the occupancy measure on $S \times A$ introduced by policy $\pi$ as $q_{\pi} \in \Delta(S \times A)$. Then we have

$$q_{\pi}(s,a) = (1-\gamma) \sum_{t=0}^{\infty} \gamma^t \mathbb{P}_{\pi}(s_t = s, a_t = a), \quad \pi(a|s) = \frac{q_{\pi}(s,a)}{\sum_{a' \in A} q_{\pi}(s, a')}.$$

Problem (M) can be reformulated as the following LSIP problem:

$$\begin{aligned}
\max_{q} \ & q^{\top} r \\
\text{s.t. } & \frac{1}{1-\gamma} q^{\top} c_y \le u_y, \ \forall y \in Y, \\
& \sum_{s',a} q(s',a)(\mathbf{1}_{\{s'=s\}} - \gamma P(s|s',a)) = (1-\gamma)\mu(s), \ \forall s \in S, \\
& q \succeq 0.
\end{aligned} \tag{4}$$

Therefore, when $M$ is already known the optimal policy $\pi^*$ can be found by solving Problem (4). And we always assume such a policy $\pi^*$ exists.

**Assumption 3.4.** Problem (M) is feasible with an optimal solution $\pi^*$, or equivalently, Problem (4) is feasible with an optimal solution $q^*$.

## 4  The SI-CRL Algorithm

In this section we present an offline reinforcement learning algorithm called SI-CRL for SICMDPs. In a high-level point of view, our algorithm is a semi-infinite version of the algorithms proposed in [21, 14]. In the first stage, SI-CRL takes an offline dataset $\{(s_i, a_i, s'_i) | i = 1, 2, \ldots, m\}$ as input and generate an empirical estimate $\widehat{P}$ of the true transition dynamic $P$. Then the algorithm constructs a confidence set (the optimistic set) according to $\widehat{P}$ that would cover the true SICMDP with high probability. Then for each policy $\pi$ we would only view its return as the largest possible return in SICMDPs in the confidence set. This method is also called the optimistic approach. In the second stage, the optimistic policy $\tilde{\pi}$ is found using a LSIP algorithm. It can be shown that the resulting policy $\tilde{\pi}$ is guaranteed to be nearly optimal, and the theoretical analysis can be found in Section 5.

Now we give a more detailed description of SI-CRL. First, the empirical estimate $\widehat{P}$ is calculated as: $\widehat{P}(s'|s,a) := \frac{n(s,a,s')}{\max(1,n(s,a))}$, where $n(s,a,s') := \sum_{i=1}^m \mathbf{1}\{s_i = s, a_i = a, s'_i = s'\}$ and $n(s,a) = \sum_{s'} n(s,a,s')$. The reason why we do not directly plug $\widehat{P}$ into Problem (4) and solve the resulting LSIP problem is due to the fact that there is no guarantee that the LSIP problem w.r.t. $\widehat{P}$ is feasible. To address this issue, we construct an optimistic set $M_\delta$ such that with high probability the true SICMDP $M$ lies in $M_\delta$. In particular, $M_\delta$ is defined via the empirical Bernstein's bound and the Hoeffding's bound [26]:

$$M_\delta := \Big\{ \langle S, A, Y, P', r, c, u, \mu, \gamma \rangle \colon |P'(s'|s,a) - \hat{P}(s'|s,a)| \le d_\delta(s,a,s'), \forall s, s' \in S, a \in A \Big\},$$

where

$$d_\delta(s,a,s') := \min \left\{ \sqrt{\frac{2\widehat{P}(s'|s,a)(1-\widehat{P}(s'|s,a))\log(4/\delta)}{n(s,a,s')}} + \frac{4\log(4/\delta)}{n(s,a,s')}, \ \sqrt{\frac{\log(2/\delta)}{2n(s,a,s')}} \right\}.$$

The next step is to solve the optimistic planning problem:

$$\max_{M' \in M_\delta, \pi} V^{\pi, M'}(\mu), \quad \text{s.t. } C^{\pi, M'}(\mu) \le u_y, \ \forall y \in Y, \tag{5}$$

where the superscript $M'$ denotes that the expectation is taken w.r.t. SICMDP $M'$.

**Theorem 4.1.** *Suppose $n \ge 3$. With probability at least $1 - 2|S|^2|A|\delta$, we have that $M \in M_\delta$, and Problem (5) is feasible.*

The proof is given in the appendix. Note that the optimization variables include both $M'$ and $\pi$, and LSIP reformulations like Problem (4) would no longer be possible. Instead, we shall introduce the state-action-state occupancy measure $z(s,a,s')$. In particular, assuming $z_{P,\pi}(s,a,s') := P(s'|s,a)q_\pi(s,a)$, we have $P(s'|s,a) = \frac{z_{P,\pi}(s,a,s')}{\sum_{x \in S} z_{P,\pi}(s,a,x)}$, and $\pi(a|s) = \frac{\sum_{s' \in S} z_{P,\pi}(s,a,s')}{\sum_{s' \in S, a' \in A} z_{P,\pi}(s,a',s')}$. Problem (5) can be reformulated as the following extended LSIP problem:

$$
\begin{aligned}
\max_z \quad & \sum_{s,a,s'} z(s,a,s')r(s,a) \\
\text{s.t.} \quad & \frac{1}{1-\gamma} \sum_{s,a,s'} z(s,a,s')c_y(s,a) \le u_y, \ \forall y \in Y, \\
& z(s,a,s') \le (\widehat{P}(s'|s,a) + d_\delta(s,a,s')) \sum_{x \in S} z(s,a,x), \forall s, s', \ a \in A, \\
& z(s,a,s') \ge (\widehat{P}(s'|s,a) - d_\delta(s,a,s')) \sum_{x \in S} z(s,a,x), \forall s, s' \in S, \ a \in A, \\
& \sum_{x \in S, b \in A} z(s,b,x) = (1-\gamma)\mu(s) + \gamma \sum_{x \in S, b \in A} z(x,b,s), \forall s \in S, \\
& z \succeq 0.
\end{aligned}
\tag{6}
$$

However, compared to LP problems, LSIP problems are typically harder to solve and there are no all-purpose LSIP solvers. Here, we choose the simple yet effective dual exchange methods [23, 32] to solve Problem 6. The SI-CRL algorithm can be summarized in Algorithm 1.

---

**Algorithm 1** SI-CRL

---

**Input:** state space $S$, action space $A$, dataset $\{(s_i, a_i, s_i')|i = 1, 2, ..., m\}$, reward function $r$, a continuum of cost function $c$, value for constraints $u$, discount factor $\gamma$

**for** each $(s, a, s')$ tuple **do**

  Set $\widehat{P}(s'|s, a) := \frac{\sum_{i=1}^m \mathbf{1}\{s_i=s, a_i=a, s_i'=s'\}}{\max\left(1, \sum_{i=1}^m \mathbf{1}\{s_i=s, a_i=a\}\right)}$

**end for**

Initialize $Y_0 = \{y_0\}$

**for** $i = 1$ **to** $T$ **do**

  Use a LP solver to solve a finite version of Problem (6) by only considering constraints in $Y_0$ and store the solution as $z_i$

  Find $y_i \approx \text{argmax}_{y \in Y} \sum_{s,a,s'} z_i(s, a, s') c_y(s, a) - u_y$

  **if** $\sum_{s,a,s'} z(s, a, s') c_{y_i}(s, a) - u_{y_i} \leq 0$ **then**

    Set $z_T = z_i$

    **BREAK**

  **end if**

  Add $y_i$ to $Y_0$

**end for**

**for** each $(s, a)$ pair **do**

  Set $\hat{\pi}(a|s) = \frac{\sum_{s'} z_T(s,a,s')}{\sum_{s',a'} z_T(s,a',s')}$

**end for**

**RETURN** $\hat{\pi}$

---

## 5 Theoretical Analysis

We give PAC-type bounds for SI-CRL under two different settings. The error of SI-CRL is decomposed into two parts: the statistical error from approximating Problem (M) with Problem (5) and the optimization error from the fact that the solution of (5) obtained by dual exchange method is inexact. On the statistical side, our goal is to determine that how many samples are required to make SI-CRL an $(\epsilon, \delta)$-optimal[1] when Problem (5) can be solved exactly, i.e., we want to find the sample complexity of SI-CRL. We show that the sample complexity of SI-CRL is $\widetilde{O}\left(\frac{|S|^2|A|^2}{\epsilon^2(1-\gamma)^3}\right)$ if the dataset we use is generated by a generative model, and $\widetilde{O}\left(\frac{|S||A|}{\nu_{\min}\epsilon^2(1-\gamma)^3}\right)$ if the dataset we use is generated by a probability measure $\nu$ defined on the space $S \times A$ and $P(\cdot|s, a)$ as considered in [11]. Here $\widetilde{O}$ means that all logarithm terms are discarded, and $\nu_{\min} := \min_{\nu(s,a)>0} \nu(s, a)$. It can be noted that the order of our sample complexity bound increases by a factor of $|S||A|$ compared to that of ordinary discounted MDP [8]. On the optimization side, we show that if the inner maximization problem w.r.t. $y$ can be solved exactly, the dual exchange method would produce an $\epsilon$-optimal solutions[2] when the number of iterations $T = O\left(\left[\text{diam}(Y)L\sqrt{|S|^2|A|d}/\epsilon\right]^d\right)$, where $L$ is the Lipschitz constant defined in Assumption 5.3. We will present our theoretical analysis in more details in the following part of this section.

### 5.1 Notation and Preliminaries

Given a stationary policy $\pi$, we define the value function $V^\pi(s) = \mathbb{E}(\sum_{t=0}^\infty \gamma^t r(s_t, a_t)|s_0 = s)$, $V^\pi = (V^\pi(s_1), \ldots, V^\pi(s_{|S|}))^\top \in \mathbb{R}^{|S|}$. Thus we have $V^\pi(\mu) = \mu^\top V^\pi$. Similarly, we define the expected cost $C_y^\pi(s) = \mathbb{E}(\sum_{t=1}^\infty \gamma^t c_y(s_t, a_t)|s_0 = s)$, $C_y^\pi = (C_y^\pi(s_1), \ldots, C_y^\pi(s_{|S|}))^\top \in \mathbb{R}^{|S|}$, thus

---

[1]The $(\epsilon, \delta)$-optimality would be defined in Definition 5.1

[2]The $\epsilon$-optimal solutions is defined in Definition 5.2

$C_y^\pi(\mu) = \mu^\top C_y^\pi$. And $\pi^*$ denotes the optimal policy. Suppose $\tilde{\pi}, \widetilde{M}$ are the solution of Problem (5) and $\widetilde{M} = \langle S, A, Y, \widetilde{P}, r, c, u, \mu, \gamma \rangle$. For a given stationary policy $\pi$, we use $\widetilde{V}^\pi(s), \widetilde{V}^\pi, \widetilde{V}^\pi(\mu)$, $\widetilde{C}_y^\pi(s), \widetilde{C}_y^\pi, \widetilde{C}_y^\pi(\mu), \tilde{q}_\pi(s,a)$, to represent the value function, expected cost, occupancy measure of SICMDP $\widetilde{M}$, respectively. We say an offline dataset $\{(s_i, a_i, s_i')|i = 1, 2, \ldots, m\}$ to be generated by a generative model if we sample according to $P(\cdot|s,a)$ for each $(s,a)$-pair $n = m/|S||A|$ times and record the results in the dataset. We say an offline dataset to be generated by probability measure $\nu$ and $P(\cdot|s,a)$ if $(s_i, a_i) \overset{i.i.d.}{\sim} \nu$ and $s_i' \sim P(\cdot|s_i, a_i)$.

An $(\epsilon, \delta)$-optimal policy is defined as follows.

**Definition 5.1.** An RL algorithm is called $(\epsilon, \delta)$-optimal for $\epsilon, \delta > 0$ if with probability at least $1 - \delta$ it can return a policy $\pi$ such that

$$V^{\pi^*}(\mu) - V^\pi(\mu) \le \epsilon; \quad C_y^\pi(\mu) - u_y \le \epsilon, \forall y \in Y.$$

An $\epsilon$-optimal solution of Problem (5) is defined as

**Definition 5.2.** A stationary policy $\hat{\pi}$ is called an $\epsilon$-optimal solution of Problem (5) for $\epsilon > 0$ if

$$|V^{\hat{\pi}}(\mu) - V^{\tilde{\pi}}(\mu)| \le \epsilon; \quad |C_y^{\hat{\pi}}(\mu) - u_y| \le \epsilon, \forall y \in Y$$

hold simultaneously.

Unless otherwise specified, we assume that $\forall (s,a) \in S \times A$, $c_y(s,a)$ is $L$-Lipschitz in $y$ w.r.t. $\|\cdot\|_2$. We also assume that $u_y$ is $L$-Lipschitz in $y$ w.r.t. $\|\cdot\|_2$. The assumptions can be formally stated as:

**Assumption 5.3.** $c_y(s,a)$ and $u_y$ are Lipschitz in $y$ w.r.t. $\|\cdot\|_2$, i.e., $\exists L > 0$ s.t. $\forall y, y' \in Y, (s,a) \in S \times A, |c_y(s,a) - c_{y'}(s,a)| \le L\|y - y'\|_2, |u_y - u_{y'}| \le L\|y - y'\|_2$.

The Lipschitz assumption is usually necessary when dealing with a semi-infinitely constrained problem [36, 23]. And this assumption is indeed quite mild because $Y$ is a compact set.

## 5.2 Sample Complexity of SI-CRL

We consider the case where the offline dataset we use is generated by a generative model. First we consider a restricted setting as in [26] where for each $(s,a)$-pair in the true SICMDP there are at most two possible next-states and provide the sample complexity bound. Then we will drop Assumption 5.4 using the same strategy as in [26] and derive the sample complexity bound of the general case. The proof can be found in the appendix.

**Assumption 5.4.** The true unknown SICMDP $M$ satisfies $P(s'|s,a) = 0$ for all but two $s' \in S$ denoted as $sa^+$ and $sa^- \in S$.

**Theorem 5.5.** *Suppose Assumption 5.4 holds, and the dataset we use is generated by a generative model with $m/|S||A| = n > \max\left\{\frac{36\log 4/\delta}{(1-\gamma)^2}, \frac{4\log 4/\delta}{(1-\gamma)^3}\right\}$. Then with probability $1 - 2|S|^2|A|\delta$, we have that*

$$V^{\pi^*}(\mu) - V^{\tilde{\pi}}(\mu) \le 24\sqrt{\frac{\log 4/\delta}{n(1-\gamma)^3}}; \quad C_y^{\tilde{\pi}}(\mu) - u_y \le 12\sqrt{\frac{\log 4/\delta}{n(1-\gamma)^3}}, \forall y \in Y.$$

**Theorem 5.6.** *Suppose Assumption 5.4 holds, the dataset we use is generated by a generative model and Problem 5 can be solved exactly. Then when $m = O\left(\frac{|S||A|\log\left(8|S|^2|A|/\delta\right)}{\epsilon^2(1-\gamma)^3}\right)$, SI-CRL is $(\epsilon, \delta)$-optimal.*

**Theorem 5.7.** *Suppose the dataset we use is generated by a generative model and Problem 5 can be solved exactly. Then when $m = O\left(\frac{|S|^2|A|^2(\log|S|)^3\log\left(8|S|^4|A|^3/\delta\right)}{\epsilon^2(1-\gamma)^3}\right)$, a modification of SI-CRL is $(\epsilon, \delta)$-optimal.*

*Remark* 5.8. Our proof strategy is similar to [26]. However, to get a $\widetilde{O}((1-\gamma)^{-3})$ bound [26] uses a tedious recursion argument. We greatly simplify the proof and achieve improvements in log terms (by a factor of $(\log(\frac{|S|}{\epsilon(1-\gamma)}))^2$) using sharper bounds on local variances of MDPs developed in [3].

*Remark* 5.9. Although Assumption 5.4 seems quite restrictive, we argue that it is necessary to establish sharp sample complexity bound, as shown in [26]. Specifically, without this assumption the "quasi-Bernstein bound" (Lemma B.4) will not hold, thus we may not be able to get the $\widetilde{O}((1-\gamma)^{-3})$ bound.

*Remark* 5.10. It can be noted that our sample complexity bound does not rely on the constraint set $Y$. This is because we consider the setting where $r$ and $c_y$ are known deterministic functions and the only source of randomness comes from estimating the unknown transition dynamic using an offline dataset.

Now we generalize our results to the case where the offline dataset is generated by a probability measure. The proof can be found in the appendix.

**Theorem 5.11.** *Suppose the dataset we use is generated by probability measure $\nu$ and Problem 5 can be solved exactly. Then when $m = O\left( \frac{|S||A|(\log |S|)^3 \log\left(8|S|^4|A|^3/\delta\right)}{\nu_{\min}\epsilon^2(1-\gamma)^3} \right)$, a modification of SI-CRL is $(\epsilon, \delta)$-optimal.*

*Remark* 5.12. Here "a modification of SI-CRL" stands for the following procedure: first we transform the original SICMDP to a new SICMDP satisfying Assumption 5.4, then we run SI-CRL to solve the new SICMDP. One may refer to the proof in Appendix B for more details.

### 5.3 Iteration Complexity of SI-CRL

We give the iteration complexity of SI-CRL, i.e., how many iterations are required to output an $\epsilon$-optimal solution of Problem (5) when the inner-loop problem can be solved exactly. Our results is a corollary of Theorem 4 in [23].

**Theorem 5.13.** *If the inner-loop maximization problem in SI-CRL can be exactly solved, then SI-CRL will output an $\epsilon$-optimal solution of Problem (5) if the number of iterations $T = O\left( \left\{ \mathrm{diam}(Y)L\sqrt{|S|^2|A|d}/[(1-\gamma)\epsilon] \right\}^d \right)$.*

## 6 Numerical Experiments

We design two numerical examples: toy SICMDP and discharge of sewage. By a set of numerical experiments, we illustrate the SICMDP model and validate the efficacy of the SI-CRL algorithm as well as the correctness of our theoretical results. We highlight that in the example of discharge of sewage we find that the SICMDP framework greatly outperforms the CMDP baseline obtained by discretizing the original problem in modeling realistic reinforcement learning problems. We implement our methods with Python and LP problems are solved using a full-featured university version of Gurobi [19]. Details of our implementation can be found in the appendix. All the experiments are run on a workstation with 8 CPUs and no GPU.

### 6.1 Toy SICMDP

We consider a most simple SICMDP with $|S| = 2$, $|A| = 2$ and $Y = [0, 1]$. Its MDP part is specified in Figure 1, where $p \in (0.5, 1)$ and $\tau \ll 1$ is a small positive number. For each $\gamma \in (0, 1)$, we design Lipschitz $c_y$ and $u_y$ such that the optimal policy takes $a_0$ with probability 0.9 and 0.5 on $s^0$ and $s^1$, respectively. For details of the construction of Toy SICMDP, one may refer to the appendix. To make our numerical results more reliable, we repeat all experiments in this subsection for 30 times and report the average results. First, we would like to check the efficacy of the SI-CRL algorithm. We set $T$ sufficiently large such that the algorithm is guaranteed to converge. Then we gradually increase $m$, the size of the dataset, and see whether SI-CRL can recover the pre-defined optimal policy. The results are shown in Figure 3. It can be noticed that as $m$ gets larger, the error term converges to zero, showing that our SI-CRL algorithm may effectively solve reinforcement learning problems for SICMDPs. Second, we would like to validate the theoretical results in Section 5. Specifically, we investigate the sample complexity of SI-CRL for a fixed $(\epsilon, \delta)$ (See Definition 5.1) when $\gamma$ and $\nu_{\min}$ vary. $T$ is set to be sufficiently large as in the previous experiment. We present the results in Figure 3. The logarithm of sample complexity vs. the transformed parameter of interest is approximately linear with slope 1, which indicates our sample complexity bounds are correct and tight.

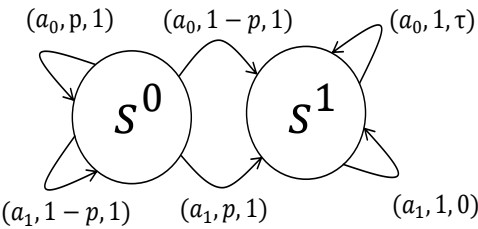

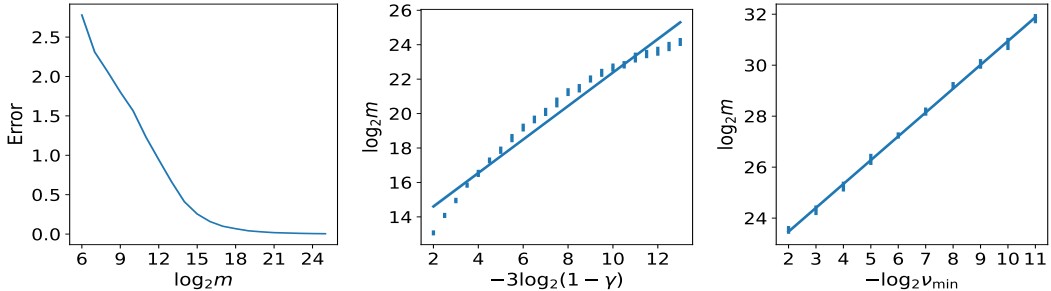

Figure 1: MDP part of Toy SICMDP: The triple means (action, probability, reward). The agent should always take action $a_0$ in both states if it sets aside the constraints.

Figure 2: (Discharge of Sewage) The satellite image is from NASA and is only for illustrative purposes. The icons represent the locations of the sewage outfalls.

Figure 3: (Toy SICMDP) Left: The policy returned by SI-CRL converges to the optimal solution as the dataset gets larger. The error term is defined as $\max\{V^{\pi^*}(\mu) - V^{\hat{\pi}}(\mu), \sup_{y \in Y} C_y^{\hat{\pi}}(\mu) - u_y\}$, the dataset is generated by generative models. Middle: Sample complexity of SI-CRL with varying $\gamma$; the dataset is generated by generative models. Right: Sample complexity of SI-CRL with varying $\nu_{\min}$; the dataset is generated by a probability measure. Here we set $\epsilon = 0.01, \delta = \frac{0.005}{|S|^2|A|}$. Straight lines are obtained by linear regression.

## 6.2 Discharge of Sewage

To demonstrate the power of the SICMDP model and the SI-CRL algorithm, we consider a more realistic and complex problem adapted from [18]. Assume there are $|S|$ sewage outfalls in a region $[0,1]^d$, with $d = 2$ or $3$, and at each time point only one single outfall is active. The active outfall would cause pollution in nearby areas, and the impact would decrease with Euclidean distance. We need to figure out a strategy to switch between neighboring outfalls to avoid over-pollution at each location of the region while minimizing the switching cost. Clearly, this problem can be formulated as a SICMDP model with $Y = [0,1]^d$ and corresponding $c_y$ and $u_y$. For details of the construction of the Discharge of Sewage, one may refer to the appendix. In the following numerical experiments, we assume that an offline dataset generated by a generative model is available.

First, we numerically validate our theoretical bounds on sample complexity and iteration complexity. In particular, we investigate the relationship between the sample complexity and iteration complexity of SI-CRL and the size of state space $|S|$. Like the case in Toy SICMDP, we find the numerical results fit well with our theoretical analysis. We show the results in Figures 4. As before, we run each experiment for 30 times and report the averaged results.

Second, we compare our method with a naive CMDP baseline 3.3, showing the advantage of SICMDP in modeling problems like Example 3.1, 3.2. In the baseline method, we only consider the constraints on a grid of $Y$ containing $T_{\text{baseline}}$ points, which allows us to model Discharge of Sewage as a standard CMDP problem with $T_{\text{baseline}}$ constraints. The CMDP problem is then solved by the algorithm proposed in [14]. We visualize the quality of solutions of our proposed method and baseline method in Figure 5. It can be found that when $T = T_{\text{baseline}}$, the policy obtained by our proposed methods is of far better quality than the policy obtained by the baseline methods.

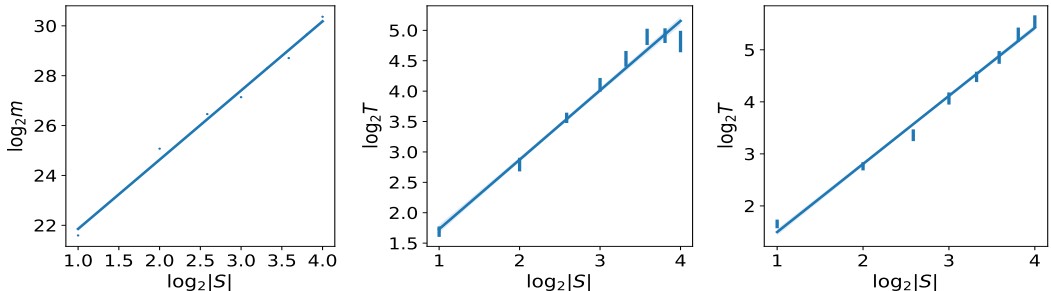

Figure 4: (Discharge of Sewage) Left: Sample complexity of SI-CRL ($\epsilon = 0.015$, $\delta = \frac{0.005}{|S|^2|A|}$, $T$ sufficiently large) with different $|S|$. Middle and right: Iteration complexity of SI-CRL ($\epsilon = 0.015$, $\delta = \frac{0.005}{|S|^2|A|}$, $m$ sufficiently large) with different $|S|$ when $d = 2$ (middle) and $d = 3$ (right), respectively. Straight lines are obtained by linear regression.

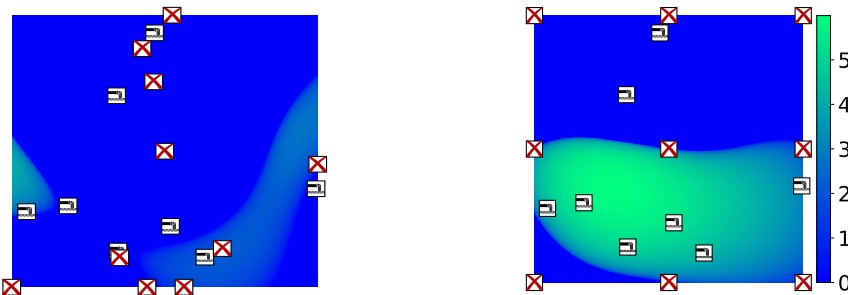

Figure 5: (Discharge of Sewage) Visualization of violation of constraints using SI-CRL (left) and baseline (right). The heat refers to the number $\log\big((C_y^{\hat{\pi}}(\mu) - u_y)_+ + 5 \times 10^{-6}\big) - \log(5 \times 10^{-6})$. Larger number means more serious violation of constraints. The red cross icons represent the $T = T_{\text{baseline}} = 9$ check points selected by the algorithms.

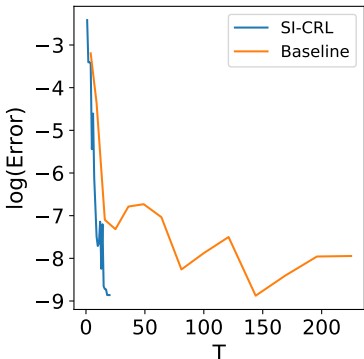

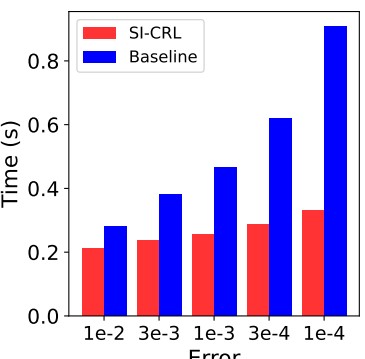

Figure 6: (Discharge of Sewage) Error term of our proposed method and the baseline method when $T$ and $T_{\text{baseline}}$ vary. ($\delta = \frac{0.005}{|S|^2|A|}$, $m$ sufficiently large)

Figure 7: Time consumption of our method and the CMDP baseline to get a solution of given accuracy. ($\delta = \frac{0.005}{|S|^2|A|}$, $m$ sufficiently large)

An anti-intuitive phenomenon is that although in our method we need to deal with multiple LP problems while in the baseline we only solve one single LP problem, our method is still more time-efficient than the CMDP baseline. Figure 7 indicates that our method takes less time to get a solution of given accuracy, which is evaluated by the error term $\sup_{y \in Y} C_y^{\hat{\pi}}(\mu) - u_y$. The reason is that in SI-CRL we solve LP problems with a dual simplex method, thus re-optimization after adding

a new constraint is much faster than re-solving the LP problem from scratch[25]. And our method needs far fewer active constraints to attain the same accuracy as the baseline methods, see Figure 6.

## 7 Conclusion

We have studied a novel generalization of CMDP that we have called SICMDP. In particular, we have considered a continuum of constraints rather than a finite number of constraints. We have devised a reinforcement learning algorithm SI-CRL to solve SICMDP problems. Furthermore, we have presented theoretical analysis for SI-CRL, establishing the sample complexity bounds as well as the iteration complexity bounds. We have also performed the extensive numerical experiments to show the efficacy of our proposed method and its advantage over traditional CMDPs. However, the SI-CRL algorithm can only handle the tabular case, with a nice offline dataset avaliable. We would study the SICMDP beyond the tabular case and develop efficient algorithms in future works.

## Acknowledgments and Disclosure of Funding

This work has been supported by the National Key Research and Development Project of China (No. 2020AAA0104400). Also, the authors would like to thank Mr. Hao Jin for helpful discussions.

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
