# A Omitted Proofs in Section 4

*Proof of Theorem 4.1.* By Lemma C.1,

$$\mathbb{P}\left(|P(s'|s,a) - \hat{P}(s'|s,a)| \leq \sqrt{\frac{2\hat{P}(s'|s,a)(1 - \hat{P}(s'|s,a))\log 4/\delta}{n}} + \frac{4\log 4/\delta}{n}\right) \geq 1 - \delta.$$

By Lemma C.2,

$$\mathbb{P}\left(|P(s'|s,a) - \hat{P}(s'|s,a)| \leq \sqrt{\frac{\log 2/\delta}{2n}}\right) \geq 1 - \delta.$$

Combining the two inequalities in a union bound, we have:

$$\mathbb{P}\left(|P(s'|s,a) - \hat{P}(s'|s,a)| \leq d_\delta(s,a,s')\right) \geq 1 - 2\delta.$$

Again we apply the union bound argument to get:

$$\mathbb{P}(M \in M_\delta) = \mathbb{P}\left(|P(s'|s,a) - \hat{P}(s'|s,a)| \leq d_\delta(s,a,s'), \forall s, s' \in S, a \in A\right) \geq 1 - 2|S|^2|A|\delta.$$

Finally, Problem (5) is feasible as long as $P \in M_\delta$ because of Assumption 3.4. $\qquad\square$

# B Omitted Proofs in Section 5

First we define some additional notations. We use $Q^\pi(s,a) := \mathbb{E}(\sum_{t=0}^\infty \gamma^t r(s_t, a))$ to denote the state-action value function. The local variance is defined as $\mathrm{Var}_P(V^\pi)(s,a) = \mathbb{E}_{s' \sim P(\cdot|s,a)}(V^\pi(s') - P(\cdot|s,a)V^\pi)^2$. We view $V^\pi$ as vector of length $|S|$ and $Q^\pi, r, \mathrm{Var}_P(V^\pi)$ as vectors of length $|S| \cdot |A|$. We overload notation and let $P$ also refer to a matrix of size $(|S| \cdot |A|) \times |S|$, where the entry $P_{(s,a),s'}$ is equal to $P(s'|s,a)$. We also define $P^\pi$ to be the transition matrix on state-action pairs induced by a stationary policy $\pi$, namely:

$$P^\pi_{(s,a),(s',a')} := P(s'|s,a)\pi(a'|s').$$

We use $\widetilde{Q}^\pi(s,a), \mathrm{Var}_{\widetilde{P}}(\widetilde{V}^\pi)(s,a), \widetilde{Q}^\pi, \mathrm{Var}_{\widetilde{P}}(\widetilde{V}^\pi), \widetilde{V}^\pi, \widetilde{\sigma}_\pi^2, \widetilde{P}, \widetilde{P}^\pi$ to denote the state-action value function, local variance, vector of state-action value function, vector of local variance, vector of value function, transition matrix, transition matrix on state-action pairs w.r.t. SICMDP $\widetilde{M}$, respectively.

**Lemma B.1.** *If Assumption 5.4 is true and $M \in M_\delta$, we have*

$$\left\|Q^\pi - \widetilde{Q}^\pi\right\|_\infty \leq \frac{2\gamma}{(1-\gamma)^2}\sqrt{\frac{\log 2/\delta}{2n}}$$

*Proof.* Given a stationary policy $\pi$, if Assumption 5.4 is true and $M \in M_\delta$,

$$\left\|\widetilde{P}(\cdot|s,a) - P(\cdot|s,a)\right\|_1 \leq 2\sqrt{\frac{\log 2/\delta}{2n}}, \forall s \in S, a \in A,$$

which implies

$$\left\|(P - \widetilde{P})V^\pi\right\|_\infty \leq \frac{2}{1-\gamma}\sqrt{\frac{\log 2/\delta}{2n}}.$$

Then we have

$$\begin{aligned}
\left\|Q^\pi - \widetilde{Q}^\pi\right\|_\infty &= \left\|\gamma\left(I - \gamma\widetilde{P}^\pi\right)^{-1}(P - \widetilde{P})V^\pi\right\|_\infty \\
&\leq \frac{\gamma}{1-\gamma}\left\|(P - \widetilde{P})V^\pi\right\|_\infty \\
&\leq \frac{2\gamma}{(1-\gamma)^2}\sqrt{\frac{\log 2/\delta}{2n}}
\end{aligned}$$

$\qquad\square$

**Lemma B.2.** *Given a stationary policy $\pi$, when Assumption 5.4 is true and $M \in M_\delta$, we have*

$$\mathrm{Var}_P(V^\pi) \leq 2\mathrm{Var}_{\widetilde{P}}(\widetilde{V}^\pi) + \frac{6}{(1-\gamma)^2}\sqrt{\frac{\log 2/\delta}{2n}} + \frac{8\gamma^2}{(1-\gamma)^4}\frac{\log 2/\delta}{2n}.$$

*Proof.* For simplicity of notation, we drop the dependence on $\pi$. By definition,

$$\begin{aligned}
\mathrm{Var}_P(V) &= \mathrm{Var}_P(V) - \mathrm{Var}_{\widetilde{P}}(V) + \mathrm{Var}_{\widetilde{P}}(V) \\
&= P(V)^2 - (PV)^2 - \widetilde{P}(V)^2 + (\widetilde{P}V)^2 + \mathrm{Var}_{\widetilde{P}}(V) \\
&= (P - \widetilde{P})(V)^2 - \left[(PV)^2 - (\widetilde{P}V)^2\right] + \mathrm{Var}_{\widetilde{P}}(V),
\end{aligned}$$

where $(\cdot)^2$ means element-wise squares. When Assumption 5.4 is true and $M \in M_\delta$, by Lemma B.1,

$$\left\|(P - \widetilde{P})(V)^2\right\|_\infty \leq \frac{2}{(1-\gamma)^2}\sqrt{\frac{\log 2/\delta}{2n}}$$

$$\begin{aligned}
\left\|\left[(PV)^2 - (\widetilde{P}V)^2\right]\right\|_\infty &\leq \left\|PV + \widetilde{P}V\right\|_\infty \left\|PV - \widetilde{P}V\right\|_\infty \\
&\leq \frac{2}{1-\gamma}\left\|PV - \widetilde{P}V\right\|_\infty \\
&\leq \frac{4}{(1-\gamma)^2}\sqrt{\frac{\log 2/\delta}{2n}}.
\end{aligned}$$

We also have

$$\begin{aligned}
\mathrm{Var}_{\widetilde{P}}(V) &= \mathrm{Var}_{\widetilde{P}}(V - \widetilde{V} + \widetilde{V}) \\
&\leq 2\mathrm{Var}_{\widetilde{P}}(V - \widetilde{V}) + 2\mathrm{Var}_{\widetilde{P}}(\widetilde{V}) \quad \text{(AM–GM inequality)} \\
&\leq 2\left\|V - \widetilde{V}\right\|_\infty^2 + 2\mathrm{Var}_{\widetilde{P}}(\widetilde{V}) \\
&\leq \frac{8\gamma^2}{(1-\gamma)^4}\frac{\log 2/\delta}{2n} + 2\mathrm{Var}_{\widetilde{P}}(\widetilde{V}) \quad \text{(Lemma B.1).}
\end{aligned}$$

Therefore, we can get

$$\mathrm{Var}_P(V^\pi) \leq 2\mathrm{Var}_{\widetilde{P}}(\widetilde{V^\pi}) + \frac{6}{(1-\gamma)^2}\sqrt{\frac{\log 2/\delta}{2n}} + \frac{8\gamma^2}{(1-\gamma)^4}\frac{\log 2/\delta}{2n}.$$

$\square$

**Lemma B.3.** *Let $p, \tilde{p}, \hat{p} \in [0,1]$ satisfy*

$$|p - \hat{p}| \leq \min\left\{\sqrt{\frac{2\hat{p}(1-\hat{p})\log 4/\delta}{n}} + \frac{4\log 4/\delta}{n}, \sqrt{\frac{\log 2/\delta}{2n}}\right\}$$

$$|\tilde{p} - \hat{p}| \leq \min\left\{\sqrt{\frac{2\hat{p}(1-\hat{p})\log 4/\delta}{n}} + \frac{4\log 4/\delta}{n}, \sqrt{\frac{\log 2/\delta}{2n}}\right\}.$$

*Then*

$$|p - \tilde{p}| \leq \sqrt{\frac{8p(1-p)\log 4/\delta}{n}} + 4\left(\frac{\log 4/\delta}{n}\right)^{3/4} + \frac{8\log 4/\delta}{n}$$

*Proof.* Assume WLOG that $\hat{p} \geq p$. Therefore,

$$\begin{aligned}
|p - \hat{p}| &\leq \sqrt{\frac{2p(1-p)\log 4/\delta}{n}} + \sqrt{\frac{2(\hat{p}-p)(1-p)\log 4/\delta}{n}} + \frac{4\log 4/\delta}{n} \\
&\leq \sqrt{\frac{2p(1-p)\log 4/\delta}{n}} + \sqrt{\frac{2\sqrt{\frac{\log 2/\delta}{2n}}\log 4/\delta}{n}} + \frac{4\log 4/\delta}{n} \\
&\leq \sqrt{\frac{2p(1-p)\log 4/\delta}{n}} + 2^{1/4}\left(\frac{\log 4/\delta}{n}\right)^{3/4} + \frac{4\log 4/\delta}{n}.
\end{aligned}$$

Similarly, we have

$$|\tilde{p} - \hat{p}| \le \sqrt{\frac{2p(1-p)\log 4/\delta}{n}} + 2^{1/4}\left(\frac{\log 4/\delta}{n}\right)^{3/4} + \frac{4\log 4/\delta}{n}.$$

Thus we may complete the proof using triangular inequality. □

**Lemma B.4.** *Given a stationary policy $\pi$, suppose Assumption 5.4 is true and $M \in M_\delta$, then*

$$|(P - \widetilde{P})V^\pi| \preceq \sqrt{\frac{8\text{Var}_P(V^\pi)\log 4/\delta}{n}} + \frac{4}{1-\gamma}\left(\frac{\log 4/\delta}{n}\right)^{3/4} + \frac{8\log 4/\delta}{n(1-\gamma)},$$

*where $\preceq$ means every element of LHS is less than or equal to the its counterpart in RHS.*

*Proof.* Let $p = P(sa^+|s, a), \tilde{p} = \tilde{P}(sa^+|s, a)$. Applying Lemma B.3 yields

$$|p - \tilde{p}| \le \sqrt{\frac{8p(1-p)\log 4/\delta}{n}} + 4\left(\frac{\log 4/\delta}{n}\right)^{3/4} + \frac{8\log 4/\delta}{n}.$$

Assume WLOG that $V^\pi(sa^+) \ge V^\pi(sa^-)$. Therefore we have

$$|(P(\cdot|s, a) - \tilde{P}(\cdot|s, a))^\top V^\pi| \le \sqrt{\frac{8p(1-p)\log 4/\delta}{n}}(V^\pi(sa^+) - V^\pi(sa^-)) + \frac{4}{1-\gamma}\left(\frac{\log 4/\delta}{n}\right)^{3/4}$$
$$+ \frac{8\log 4/\delta}{n(1-\gamma)}.$$

Since

$$p(1-p)(V^\pi(sa^+) - V^\pi(sa^-))^2 = [pV^\pi(sa^+)^2 + (1-p)V^\pi(sa^-)^2] - [pV^\pi(sa^+) + (1-p)V^\pi(sa^-)]^2$$
$$= \text{Var}_P(V^\pi)$$

We may get

$$|(P(\cdot|s, a) - \widetilde{P}(\cdot|s, a))^\top V^\pi| \le \sqrt{\frac{8\text{Var}_P(V^\pi)(s, a)\log 4/\delta}{n}} + \frac{4}{1-\gamma}\left(\frac{\log 4/\delta}{n}\right)^{3/4} + \frac{8\log 4/\delta}{n(1-\gamma)},$$

which completes the proof. □

**Lemma B.5.** *Given a stationary policy $\pi$, suppose Assumption 5.4 is true and $M \in M_\delta$, then we have*

$$\left\|V^\pi - \widetilde{V}^\pi\right\|_\infty \le \frac{4}{(1-\gamma)^{3/2}}\sqrt{\frac{\log 4/\delta}{n}} + \frac{4\sqrt{6}}{(1-\gamma)^2}\left(\frac{\log 4/\delta}{n}\right)^{3/4} + \frac{8}{(1-\gamma)^4}\left(\frac{\log 4/\delta}{n}\right)^{3/2}.$$

*Proof.* From Lemma C.3, Lemma C.4, Lemma C.5, Lemma B.4 and the fact that $\left(I - \gamma\widetilde{P}^\pi\right)^{-1}$ has positive entries, we know

$$\left\|Q^\pi - \widetilde{Q}^\pi\right\|_\infty = \gamma\left\|\left(I - \gamma\widetilde{P}^\pi\right)^{-1}(P - \widetilde{P})V^\pi\right\|_\infty$$

$$\le \sqrt{\frac{8\log 4/\delta}{n}}\left\|\left(I - \gamma\widetilde{P}^\pi\right)^{-1}\sqrt{\text{Var}_P(V^\pi)}\right\|_\infty + \frac{4}{(1-\gamma)^2}\left(\frac{\log 4/\delta}{n}\right)^{3/4} + \frac{8}{(1-\gamma)^2}\left(\frac{\log 4/\delta}{n}\right)$$

$$\le \sqrt{\frac{16\log 4/\delta}{n}}\left\|\left(I - \gamma\widetilde{P}^\pi\right)^{-1}\sqrt{\text{Var}_{\widetilde{P}}(\widetilde{V}^\pi)}\right\|_\infty + \frac{4\sqrt{6}}{(1-\gamma)^2}\left(\frac{\log 4/\delta}{n}\right)^{3/4} + \frac{8}{(1-\gamma)^3}\left(\frac{\log 4/\delta}{n}\right)$$

$$\le \frac{4}{(1-\gamma)^{3/2}}\sqrt{\frac{\log 4/\delta}{n}} + \frac{4\sqrt{6}}{(1-\gamma)^2}\left(\frac{\log 4/\delta}{n}\right)^{3/4} + \frac{8}{(1-\gamma)^3}\left(\frac{\log 4/\delta}{n}\right).$$

The proof is completed since $\left\|V^\pi - \widetilde{V}^\pi\right\|_\infty \le \left\|Q^\pi - \widetilde{Q}^\pi\right\|_\infty$ by definitions. □

**Lemma B.6.** *Suppose Assumption 5.4 is true and $n > \frac{6\log 4/\delta}{(1-\gamma)^{5/2}}$, then with probability at least $1 - 2|S|^2|A|\delta$, we have*

$$\left\|V^\pi - \widetilde{V}^\pi\right\|_\infty \le 12\sqrt{\frac{\log 4/\delta}{n(1-\gamma)^3}}$$

$$\left\|C^\pi - \widetilde{C}^\pi_y\right\|_\infty \le 12\sqrt{\frac{\log 4/\delta}{n(1-\gamma)^3}}, \forall y \in Y$$

*Proof.* When Assumption 5.4 is true and $M \in M_\delta$, it follows from Lemma B.5 that

$$\left\|V^\pi - \widetilde{V}^\pi\right\|_\infty \le \frac{4}{(1-\gamma)^{3/2}}\sqrt{\frac{\log 4/\delta}{n}} + \frac{4\sqrt{6}}{(1-\gamma)^2}\left(\frac{\log 4/\delta}{n}\right)^{3/4} + \frac{8}{(1-\gamma)^3}\left(\frac{\log 4/\delta}{n}\right).$$

And by setting $n > \max\left\{\frac{36\log 4/\delta}{(1-\gamma)^2}, \frac{4\log 4/\delta}{(1-\gamma)^3}\right\}$ we will get

$$\left\|V^\pi - \widetilde{V}^\pi\right\|_\infty \le 12\sqrt{\frac{\log 4/\delta}{n(1-\gamma)^3}}.$$

Similar arguments can be applied to bound $\left\|C^\pi_y - \widetilde{C}^\pi_y\right\|_\infty$. Since by Theorem 4.1 we have

$$\mathbb{P}(M \in M_\delta) \ge 1 - 2|S|^2|A|\delta,$$

the proof is completed. $\qquad\square$

*Proof of Theorem 5.5.* By Lemma B.5, we know that with probability $1 - 2|S|^2|A|\delta$,

$$\left\|V^{\tilde\pi} - \tilde{V}^{\tilde\pi}\right\|_\infty \le 12\sqrt{\frac{\log 4/\delta}{n(1-\gamma)^3}}$$

$$\left\|V^{\pi^*} - \tilde{V}^{\pi^*}\right\|_\infty \le 12\sqrt{\frac{\log 4/\delta}{n(1-\gamma)^3}}.$$

Thus

$$|V^{\tilde\pi}(\mu) - \tilde{V}^{\tilde\pi}(\mu)| \le 12\sqrt{\frac{\log 4/\delta}{n(1-\gamma)^3}}$$

$$|V^{\pi^*}(\mu) - \tilde{V}^{\pi^*}(\mu)| \le 12\sqrt{\frac{\log 4/\delta}{n(1-\gamma)^3}}.$$

Noting that $\tilde{V}^{\tilde\pi}(\mu) \ge \tilde{V}^{\pi^*}(\mu)$, we may get

$$V^{\pi^*}(\mu) - V^{\tilde\pi}(\mu) \le V^{\pi^*}(\mu) - \tilde{V}^{\pi^*}(\mu) + \tilde{V}^{\tilde\pi}(\mu) - V^{\tilde\pi}(\mu)$$

$$\le |V^{\pi^*}(\mu) - \tilde{V}^{\pi^*}(\mu)| + |\tilde{V}^{\tilde\pi}(\mu) - V^{\tilde\pi}(\mu)|$$

$$\le 24\sqrt{\frac{\log 4/\delta}{n(1-\gamma)^3}}.$$

Similarly, when

$$|C^{\tilde\pi}_y(\mu) - \tilde{C}^{\tilde\pi}_y(\mu)| \le 12\sqrt{\frac{\log 4/\delta}{n(1-\gamma)^3}}, \forall y \in Y,$$

we may get

$$C^{\tilde\pi}_y(\mu) - u_y \le 12\sqrt{\frac{\log 4/\delta}{n(1-\gamma)^3}}, \forall y \in Y.$$

since $\tilde{C}^{\tilde\pi}_y(\mu) \le u_y$. $\qquad\square$

*Proof of Theorem 5.6.* Theorem 5.6 is a direct corollary of Theorem 5.5. □

*Proof of Theorem 5.7.* The proof is nearly identical to the proof of Theorem 3 in [26]. The idea is to augment each state/action pair of the original mdp with $|S| - 2$ states in the form of a binary tree as pictured in the diagram below.

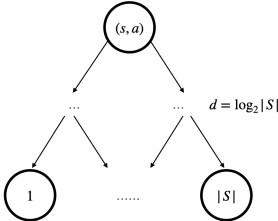

The intention of the tree is to construct an SICMDP, $\bar{M} = \langle \bar{S}, A, Y, \bar{P}, \bar{r}, \bar{c}, u, \mu, \bar{\gamma} \rangle$ that with appropriate transition probabilities is functionally equivalent to $M$ while satisfying Assumption 5.4. The rewards and costs in the added states are set to zero. Since the tree has depth $d = O(\log_2 |S|)$, it now takes $d$ time-steps in the augmented SICMDP to change states once in the original SICMDP. Therefore we must also rescale the discount factor $\bar{\gamma}$ by setting $\bar{\gamma} < \gamma^d$. Now we have

$$|\bar{S}| = O(|S|^2|A|)$$
$$\frac{1}{1 - \bar{\gamma}} = \frac{\log |S|}{1 - \gamma}.$$

Then we complete the proof by applying results in Theorem 5.6. □

*Proof of Theorem 5.11.* By Theorem C.6, we have for any fixed $(s, a) \in S \times A$

$$\mathbb{P}(n(s, a) < m\nu_{\min}/2) \leq \mathbb{P}(n(s, a) < m\nu(s, a)/2)$$
$$\leq e^{-m\nu(s,a)} \left( \frac{em\nu(s,a)}{em\nu(s,a)/2} \right)^{\frac{m\nu(s,a)}{2}}$$
$$= \left( \sqrt{\frac{e}{2}} \right)^{-\nu(s,a)m}$$
$$\leq \left( \sqrt{\frac{e}{2}} \right)^{-\nu_{\min}m}$$

□

Let $m = \frac{2}{1 - \log 2} \frac{\log 2|S||A|/\delta}{\nu_{\min}}$, we have $\mathbb{P}(n(s, a) \geq m\nu_{\min})) \geq 1 - \delta/2|S||A|$. Therefore, with probability at least $1 - \delta/2$, we can get

$$n(s, a) > m\nu_{\min}, \forall (s, a) \in S \times A.$$

Then our problem is reduced to the case that the offline dataset is generated by generative models. The proof is completed by using results in Theorem 5.7.

*Proof of Theorem 5.13.* See Theorem 4 in [23]. □

## C  Auxiliary Lemmas

**Lemma C.1** (Empirical Bernstein Inequality). *Suppose $n \geq 3$, $\{X_1, ..., X_n\}$ be $n$ i.i.d. random variables with values in $[0, 1]$. Let $\delta > 0$. Then with probability at least $1 - \delta$ we have*

$$\left| \mathbb{E}X_1 - \frac{\sum_{i=1}^n X_i}{n} \right| \leq \sqrt{\frac{2\mathbb{V}_n(X_{1:n}) \log 4/\delta}{n}} + \frac{4 \log 4/\delta}{n},$$

where $\mathbb{V}_n(X_{1:n}) := \frac{1}{n(n-1)} \sum_{i,j} \frac{(X_i - X_j)^2}{2}$ denotes the empirical variance of the dataset $\{X_1, ..., X_n\}$.

*Proof.* See Theorem 11 in [28]. □

**Lemma C.2** (Hoeffding's Inequality). *Suppose $\{X_1, ..., X_n\}$ be $n$ i.i.d. random variables with values in $[0, 1]$. Let $\delta > 0$. Then with probability at least $1 - \delta$ we have*

$$\left| \mathbb{E} X_1 - \frac{\sum_{i=1}^n X_i}{n} \right| \leq \sqrt{\frac{\log 2/\delta}{2n}}$$

*Proof.* See Theorem 2.2.6 in [38]. □

**Lemma C.3.** *For any policy $\pi$ and transition probabilities $P, \widetilde{P}$, we have that*

$$Q^\pi - \widetilde{Q}^\pi = \gamma \left( I - \gamma \widetilde{P}^\pi \right)^{-1} (P - \widetilde{P}) V^\pi$$

*Proof.* See Lemma 2 in [3]. □

**Lemma C.4.** *For any policy $\pi$, any transition probability $P$ and any vector $v \in \mathbb{R}^{|S| \cdot |A|}$, we have*

$$\left\| (I - \gamma P^\pi)^{-1} v \right\|_\infty \leq \|v\|_\infty / (1 - \gamma).$$

*Proof.* See Lemma 3 in [3]. □

**Lemma C.5.** *For any policy $\pi$ and any transition probability $P$, we have*

$$\left\| (I - \gamma P^\pi)^{-1} \sqrt{\mathrm{Var}_P^\pi} \right\|_\infty \leq \sqrt{\frac{2}{(1 - \gamma)^3}},$$

*where $\sqrt{\cdot}$ is defined as the element-wise square root.*

*Proof.* See Lemma 4 in [3]. □

**Theorem C.6** (Chernoff's Inequality). *Let $X_i$ be independent Bernoulli random variables with parameter $p_i$. Consider their sum $S_N = \sum_{i=1}^N X_i$ and denote its mean by $\mu = \mathbb{E} S_N$. Then, for any $t < \mu$, we have*

$$\mathbb{P}(S_N < t) \leq e^{-\mu} \left( \frac{e\mu}{t} \right)^t.$$

*Proof.* See [38]. □

## D   Details of Numerical Experiments

The code of our algorithms and construction of the corresponding environments has been released on https://github.com/pengyang7881187/SICMDP.

### D.1   Construction of Toy SICMDP

Without loss of generality, assume $Y = [0, 3]$. We split $Y = [0, 3]$ to $Y_1 = [0, 1]$, $Y_2 = [1, 2]$ and $Y_3 = [2, 3]$. Intuitively, in $Y_1$ ($Y_3$), we restrict the agent to take action $a_0$ in $s^0$ ($s^1$) with the given probability from the target policy $\tilde{\pi}$. (Recall, taking action $a_0$ is always better if we set aside the constraints.) We introduce $Y_2$ to obtain $L$-Lipschitz of $c_y$ and $u_y$. Assume $f : [-0.5, 0.5] \to \mathbb{R}$ is a continuous-differentiable even function, with unique maximum point $0$ and zero point $0.5$, we use $f(x) = (1 + \cos(2\pi x)) \cos(2\pi x)$ in practice. Let $c_y(s^0, a_0) = f(y - 0.5)$ for $y \in Y_1$, $c_y(s^1, a_0) = f(y - 2.5)$ for $y \in Y_3$, and $c \equiv 0$ otherwise. For each $\gamma \in (0, 1)$, we define $u^1 = C_{0.5}^{\tilde{\pi}} > 0$ and $u^1 = C_{2.5}^{\tilde{\pi}} > 0$. Let $u_y \equiv u^1$ in $Y_1$, $u_y \equiv u^2$ in $Y_3$ and we make linear interpolate $u_y$ in $Y_2$.

So far, we have constructed Lipschitz $c_y$ and $u_y$. The only active constraints are $C_{0.5}^\pi \leq u^1$ and $C_{2.5}^\pi \leq u^2$ and the optimal policy is the known target policy $\tilde{\pi}$.

### D.2 Construction of Discharge of Sewage

In the numerical experiments, we generate this environment randomly. In each experiment, we sample the environments 30 times and report the average result.

Positions of sewage outfalls, transition dynamics and rewards are sampled uniformly on $Y$, probability simplex and $[0, 1]$ respectively.

Assume $f : [0, +\infty) \to [0, +\infty)$ is a continuous-differentiable decreasing function, we use $f(x) = \frac{1}{1+x^2}$ in practice. Let $c_y(s, a) = c_y(s) = f(\|y - s\|_2)$, where $s$ also represents the position of the state (outfall).

Given a target state-occupancy measure $d$ which can be generated randomly or specified in advance, we define $u_y = (1 + \Delta) \sum_{s \in S} d(s) c_y(s)$, where $\Delta$ is a small positive number. The feasibility of the resulting SICMDP is not guaranteed even if $\Delta = 0$, we reject those infeasible environments and re-sample. The SICMDP would be nontrivial if we choose a suitable $\Delta$.

### D.3 Implementation of SICMDP

The only thing we need to clarify is how to solve the maximization problem to generate new $y_i$ in $i$th iteration. Since the cost function $c_y(s, a)$ can be non-convex in $y$, we choose to solve the maximization problem by brute force. Specifically, we first create a grid of $Y$ of size $10^5$, and then find the grid point with max objective. This method works well since in the problems we consider $Y$ is of low dimensions.