# OpenReview forum: "Semi-infinitely Constrained Markov Decision Processes"
_NeurIPS.cc/2022/Conference — NeurIPS 2022 Accept_

### Official Review · Reviewer_KTCv · 2022-07-10

**Rating:** 7
**Confidence:** 3
**Soundness:** 3 good
**Presentation:** 3 good
**Contribution:** 3 good

**Summary:**

This paper proposes a generalization for the Constrained MDPs, coined as semi-infinitely constrained MDPs. This framework extends the general CMDP making it able to handle a continuum of constraints. They transform the semi-infinite CMDP into a linear semi-infinite programming problem and then apply tools from semi-infinite programming literature to solve it. They present solid theoretical derivation and supports the claims with thorough empirical analysis.

**Questions:**

The paper is carefully written answering most questions, thanks to the authors.

Line 128 refers to "the optimistic approach". But this was never introduced in the paper before, right? Please note what it is in the paper before referring.

Algorithm 1 incrementally adds new constraints from the set of constraints. Is this set of constraints discrete, not continuous? If yes, then is the claim of incorporating continuum of constraints correct? Also how it differs from the brute-force approach of discretizing a continuous set of constraints and the using the general CMDP framework? On the other hand, if the set of constraints is continuous, is it possible to clarify the process a bit on how the incremental constraint incorporation works in algorithm 1?

**Ethics Review Area:**

["I don’t know"]

**Limitations:**

Yes

**Strengths And Weaknesses:**

The paper is well organized and well written proposing a novel and important direction of incorporating a continuum of constraints into the CMDP framework. The theories and the algorithm are presented clearly with convincing empirical results.

Examples 3.1 and 3.2 presented in the paper are sound and makes the motivation of the proposed SICMDP framework pretty clear.

The SI-CRL algorithm presented in section 4 is simple, well organized and easy to follow.

The theoretical analysis presented in section 5 is rigorous. The authors provide both the computational complexity and sample complexity for the proposed algorithm.

Section 6 provides a detailed experimental analysis of SI-CRL algorithm on one toy  and one more realistic problem domain. The results clearly show that SI-CRL outperforms the baseline method both in terms of reducing constraint violations and minimizing the errors.

---

> ### Author Response · Authors · 2022-07-28
> **Response to reviewer KTCv**
>
> We greatly appreciate your efforts as well as the positive feedback. We list our responses to your comments/questions below.
>
> * Missing definition of the optimistic approach:
> Thanks for your advice.
> We agree that it is important to make our paper self-contained, and would add some explanations about the optimistic approach in revision.
>
> * About the discretization in Algorithm 1:
> This is really a good question.
> And we would like to elaborate on the difference between Algorithm 1 and the naive discretization method and explain why the SIP-based Algorithm 1 is better.
> Both Algorithm 1 and the naive discretization method use a finite-constraint problem as a surrogate to approximate the original problem.
> To be concise, the difference between the two algorithms is that Algorithm 1 discretizes the original problem in a smarter way.
> Note that our Algorithm 1 incrementally adds the most violated constraint in each iteration step to the constraint set of the surrogate problem.
> In contrast, the naive discretization method simply forms a grid of the constraint set $Y$ and naively adds all grid points to the constraint set of the surrogate problem.
> Our numerical experiments (please see Figure 6) show that to get the same performance the naive method would require us to consider much more constraints in the surrogate problem, resulting in computational inefficiency (please see Figure 7).
> The reason that Algorithm 1 is better is Algorithm 1 takes advantage of the special structure of the SICMDP problem (the constraint function relies on $y$ in a continuous way) while the naive discretization does not.
>
>
> We hope our response is satisfactory.

---

### Official Review · Reviewer_iCiQ · 2022-07-12

**Rating:** 6
**Confidence:** 5
**Soundness:** 3 good
**Presentation:** 3 good
**Contribution:** 3 good

**Summary:**

The paper "Semi-Infinitely Constrained Markov Decision Processes" proposes a generalization of CMDPs from finite constraints to a continuum of constraints. The authors devise a generative model based reinforcement learning algorithm for Semi-Infinitely Constrained Markov Decision Processes", and characterized the sample complexity. Numerical experiments are provided to validate the performance of algorithm.

**Questions:**

Remark 5.10: The sample complexity bounds do not seem to depend on $Y$. Will this result translate to classical CMDP case?

Because, existing bounds in the classical CMDP seem to suggest the sample complexity increases with the number of constraints.


The details on the experiments are missing. How is the LISP being solved?

**Limitations:**

Yes, this algorithm works for only tabular case, and only when a offline data set is provided apriori.

**Strengths And Weaknesses:**

The paper is reasonably well written, presentation easy to follow. The use cases in Section $3$ illustrate the applicability of this approach.

The theoretical results (sample complexity bounds) are strong. But the novelty in the analysis seems to be limited, since multiple works have addressed similar problems (classical CMDPs) in the past, and this approach is very similar to the earlier works. [HasanzadeZonuzy, 2021].

The algorithm is a generative model based, it only works with a data provided apriori.

Numerical experiments indicate the advantage of the algorithm w.r.t the baseline, but this section is weak.

---

> ### Author Response · Authors · 2022-07-28
> **Response to Reviewer iCiQ**
>
> We greatly appreciate your efforts as well as the positive feedback. We list our responses to your comments/questions below.
>
> * Relationship between our bounds and $Y$: Our sample complexity bound does not depend on the size of $Y$ and it also applies to the case of finite CMDP.
>     Recall that for arbitrary reward function $r(s,a)\in[0,1]$, as long as the true transition probability is contained in the optimistic confidence set $M_\delta$ we can show the bound like
>     $$
>     ||V^\pi-\tilde V^\pi||_\infty\leq K\left(\frac{1}{1-\gamma}, |S|,|A|,\delta,n\right),
>     $$
>     where $K$ stands for an upper bound function.
>     Since we assume the cost function $c_y(s,a)\in[0,1]$ is known a priori, the bounds $||C_y-\tilde{C_y}||_\infty\leq K\left(\frac{1}{1-\gamma}, |S|,|A|,\delta,n\right)$ also holds for any $y$ as long as the confidence set contains the true transition probability (please see Lemma B.4, Lemma B.5).
>     Therefore, the sample complexity bounds should have nothing to do with the set $Y$.
>     In addition, our iteration complexity bounds (corresponding to the error from the optimization side) do depend on the size of $Y$ (please see Theorem 5.12).
>
> * Difference between the theory part of our paper and that of [HasanzadeZonuzy, 2021]: The most prominent difference is that our sample complexity bounds are independent of the set $Y$ and their sample complexity bounds depend on $N$, which is the total number of constraints.
>     We note that [HasanzadeZonuzy, 2021] also assumes the cost functions are known a priori, and we believe the reason that their bounds depend on $N$ is the use of unnecessary uniform arguments.
> In other words, we break the misconception that CMDP would become statistically harder when the number of constraints increases even if all constraints functions are known a priori.
>
> * Details of our experiment: The LSIP problem is solved by the dual-exchange method (please see Algorithm 1), and the inner maximization problem can be solved by any optimization subroutines.
>     In our experiment, the inner maximization problem is solved by brute-force searching, please see Appendix D.3 for more details.
>     In addition, the details of the construction of our numerical examples can be found in Appendix D.1 and Appendix D.2.
>
> * Extention to the online setting & non-tabular cases: In fact, we believe it is possible to use policy-optimization-type algorithms combined with function approximations (like NN) to solve the SICMDP problem in the online setting & non-tabular cases. However, we think the extension is beyond the scope of this paper and we would leave it as future work.
>
> We hope our response is satisfactory.

---

### Official Review · Reviewer_V6h8 · 2022-07-12

**Rating:** 4
**Confidence:** 4
**Soundness:** 3 good
**Presentation:** 3 good
**Contribution:** 2 fair

**Summary:**

This paper generalizes the traditional constrained Markov decision process (CMDP) to a CMDP that has infinitely many constraints, which is called semi-infinitely CMDP (SICMDP). The main challenge of solving the optimal policy for such a SICMDP is to deal with a continuum of constraints, and existing constrained reinforcement learning (CRL) methods do not work anymore. In this paper, the authors propose the first RL algorithm for the tabular SICMDP based on linear semi-infinitely programming (LSIP). Specifically, they first construct a set of dynamics under the offline setting or the generative model setting, and formulate an optimistic planning problem. Then, they augment the state-action occupancy measure to the state-action-state occupancy measure, by which they transform the optimistic planning problem into LSIP. Finally, such LSIP is solved via an existing method. Furthermore, the authors prove the sample complexity and iteration complexity of their algorithm under some assumptions. In the experiments, they show the effectiveness of their method and the tightness of their theoretical bounds.

**Questions:**

1. In this paper, the reward and cost functions are known a priori. Do the method and theory still work when the reward and cost functions are unknown and learned from samples as the dynamic?

2. The method in this paper only works for tabular SICMDPs with small state and action spaces. RL algorithms for SICMDPs beyond the tabular case would be significant.

3. What is the technical difference between this work and [21], except that this work is a LSIP?

4. The authors should clarify their contributions more clearly.

5. It would be interesting to design algorithms for the episodic setting in addition to the generative model setting and offline setting, since the last two setting omit exploration and are somewhat easier.


**Limitations:**

1. The PAC bound in this paper is flawed, and should consider both the statistical and optimization errors.

2. The technical difference between this work and [21] should be explained. And the authors should clarify their contributions more clearly.

3. The extensions of this paper to the episodic setting or large MDPs would be appreciated.


**Strengths And Weaknesses:**

Strengths
1. This paper generalizes CMDP to SICMDP, which is a novel model that has many applications.
2. This paper introduces tools from LSIP to RL and proposes the first RL algorithm for solving the optimal policy of SICMDP.
3. This paper provides the sample complexity and iteration complexity of their algorithm.

Weaknesses
1. About solution:
- Although this paper is the first one to focus on SICMDP, the proposed solution has limited novelty. Specifically, this paper constructs a confidence interval for the dynamic and formulates an optimistic planning problem, which is widely used in model-based papers, e.g., [14] and [21].
- This paper transforms the optimistic planning problem into LSIP form. Such transform is similar to [21]. At last, the LSIP is solved by an existing solver.

2. About theory:
- The PAC bound in this paper is flawed. The authors prove the sample complexity of their algorithm under the assumption that Problem (5) can be solved exactly, which is unreasonable. The authors should prove that, in order to get a $\epsilon$-optimal policy with probability $1-\delta$, how many samples and number of iterations are needed.
- In Theorem 5.7 and 5.11, the authors say “a modification of SI-CRL is …”. But they do not describe what the modification is.

3. About experiments:
- In Discharge of Sewage domain, the authors compare the CMDP baseline obtained by discretizing the original problem. However, they only discretize the infinite set Y into 9 points, which may be unfavorable for such a CMDP baseline. They should add experiments about discretizing Y into more points.
- In Figure 5, the authors show the results of constraint violation. But they should also show the performance of the objective, i.e., the cumulative rewards, as in regular CRL papers.

4. About related work:
- The authors should explain the difference between this work and [21].
- The literature review on CRL works is far from thorough. The authors should add more related works about CRL.

5. About Writing:
- In the penultimate paragraph of Introduction, some notations (e.g., d, diam(Y)) are used 	before introducing them, which is confusing for readers.

---

> ### Author Response · Authors · 2022-07-28
> **Response to Reviewer V6h8**
>
> Thanks for your feedback and constructive advice. We list our responses to your comments/questions below.
>
> * About the "flawed PAC bound": We do consider both statistical error and optimization error in our paper. We provide the iteration complexity of Algorithm 1 in Theorem 5.12. The answer to "how many samples and numbers of iterations are needed to get a $\epsilon$-optimal policy with probability $1-\delta$" is an immediate consequence of combining Theorem 5.7 and Theorem 5.12.
>
> * About the difference between this work and  [HasanzadeZonuzy, 2021]: One of the most important differences is that our sample complexity bounds are independent of the set $Y$ and their sample complexity bounds depend on $N$, which is the total number of constraints. In other words, we break the misconception that CMDP would become statistically harder when the number of constraints increases even if all constraints functions are known a priori. And we achieve this by avoiding the use of unnecessary uniform arguments. You may also check our response to reviewer iCiQ (https://openreview.net/forum?id=ohk8bILFDkk&noteId=qrjFCcAkyXM) for further details. In addition, we provide results about controlling the optimization error of Algorithm 1, which does not appear in [HasanzadeZonuzy, 2021] since in their paper the LP problems can be solved exactly.
>
> * About discretizing $Y$ into more points: We do discretize $Y$ into more points. In fact, we discretize $Y$ into much more than nine points such that the baseline attains a given level of optimality. Please see Figure 4 (more than 32 points), Figure 6 (more than 200 points), and Figure 7. Only in Figure 5 $Y$ is discretized into nine points. However, this figure is only a minor part of our empirical study for illustrative purposes.
>
> * About showing the performance of the objective: We consider the performance of the objective in every numerical experiment. It is because our optimality level is defined as
> $$\max \left\\{V^{\pi^{*}}(\mu)-V^{\hat{\pi}}(\mu), \sup_{y \in Y} C_{y}^{\hat{\pi}}(\mu)-u_{y} \right\\}$$ (please check the caption of Figure 3).
> When we say an algorithm attains a certain level of optimality, we mean that the solution both has near-optimal objectives and nearly violates no constraints. In Figure 5 we only show the results of constraint violation for illustrative purposes.
>
> * Unknown rewards and costs: In the literature of RL theory the reward (and costs) are often assumed to be known because the main difficulty is estimating the transition dynamics. We believe our method and theory still work when the rewards and costs are unknown. A possible approach is that first we get the estimated reward $\hat r$ and cost $\hat{c_y}$. The latter can be obtained by regressing $c_{y_i}(s_i,a_i)$ against $y$ using non-parametrical regression. Then we may directly plug the estimated function into Algorithm 1. We may still get useful performance guarantees as long as $\sup_{y\in Y}\\| \hat{c_y}-c_y\\|_\infty$ can be controlled.
>
> * Extention to the online setting & non-tabular cases: In fact, we believe it is possible to use policy-optimization-type algorithms combined with function approximations (like NN) to solve the SICMDP problem in the online setting & non-tabular cases. However, we think the extension is beyond the scope of this paper and we would leave it as future work.
>
> * Clearing up our contribution: Our most significant contribution is that we propose a new constrained RL model called SICMDP, where a continuum of constraints is considered. We also find effective algorithms for solving SICMDP by first introducing the tools from SIP into the area of constrained RL. In addition, we provide theoretical guarantees of our proposed algorithm from both the statistical perspective and the optimization perspective and show its effectiveness through empirical studies.
>
> * About the modification of SI-CRL: Thanks for pointing out this. The modification procedure can be found in proof of Theorem 5.7, Appendix B. Specifically, we need to first convert the original SICMDP to a SICMDP satisfying Assumption 5.4 and run Algorithm 1 on the modified SICMDP. We would add a remark describing the specific modification procedure in revision.
>
>
> We hope our response is satisfactory.

---

> ### Author Response · Authors · 2022-08-08
> **Looking Forward to Hearing from Reviewer V6h8**
>
> Dear Reviewer V6h8,
>
> As the author-reviewer discussion period will end soon, we will greatly appreciate it if you could let us know whether our response satisfactorily addresses your concerns. If not, we are glad to give further explanations.
>
> Yours,
> Authors

---

### Meta-Review · Area_Chair_mKim · 2022-08-26

**Recommendation:** Accept
**Confidence:** Less certain

**Metareview:**

This paper develops an RL method for semi-infinitely constrained MDPs.  Apparently they are the first to do this, though the algorithmic contributions are not particularly innovative and the theoretical results are not particularly surprising.  I see this as a fine, albeit technical, contribution to the literature.  I am on the fence with regards to acceptance.

**Award:**

No

---

### Decision · Program_Chairs · 2022-09-14

Accept